**Rapid formation of intense haze episodes via aerosol-boundary layer feedback in Beijing**
Yonghong Wang[1,2], Miao Yu[3], Yuesi Wang[1,6], Guiqian Tang[1], Tao Song[1], Putian Zhou[2], Zirui Liu[1],
Bo Hu[1], Dongsheng Ji[1], Lili Wang[1], Xiaowan Zhu[1], Chao Yan[2], Mikael Ehn[2], Wenkang
Gao[1],Yuepeng Pan[1], Jinyuan Xin[1], Yang Sun[1], Veli-Matti Kerminen[2], Markku Kulmala[2,4,5] and
Tuukka Petäjä[2,4,5]
[1]State Key Laboratory of Atmospheric Boundary Layer Physics and Atmospheric Chemistry
(LAPC), Institute of Atmospheric Physics, Chinese Academy of Sciences, Beijing 100029, China
[2]Institute for Atmospheric and Earth System Research / Physics, Faculty of Science, P.O.Box 64,
00014 University of Helsinki, Helsinki, Finland
[3]Institute of Urban Meteorology, China Meteorological Administration, Beijing, China
[4]Joint international research Laboratory of Atmospheric and Earth SysTem sciences (JirLATEST),
Nanjing University, Nanjing, China
[5]Aerosol and Haze Laboratory, Beijing Advanced Innovation Center for Soft Matter Science and
Engineering, Beijing University of Chemical Technology (BUCT), Beijing, China
[6]Centre for Excellence in Atmospheric Urban Environment, Institute of Urban Environment,
Chinese Academy of Science, Xiamen, Fujian 361021, China

21           Corresponding authors: Yuesi Wang and Markku Kulmala

22           E-mail: wys@mail.iap.ac.cn; markku.kulmala@helsinki.fi



25           Revised to: Atmospheric Chemistry and Physics






**Keywords**: $PM_{2.5}$, Mixing layer height, Turbulent kinetic energy, vertical measurement, model,
feedback

**Abstract**

Although much efforts have been put on studying air pollution, our knowledge on the mechanisms of frequently occurred intense haze episodes in China is still limited. In this study, using three years of measurements of air pollutants at three different height levels on a 325-meter Beijing meteorology tower, we found that a positive aerosol-boundary layer feedback mechanism existed at three vertical observation heights during intense haze polluted periods within the mixing layer. This feedback was characterized by a higher loading of $PM_{2.5}$ with a shallower mixing layer. Modeling results indicated that the presence of $PM_{2.5}$ within boundary layer lead to reduced surface temperature, relative humidity and mixing layer height during an intensive haze episode. Measurements showed that the aerosol-boundary layer feedback was related to the decrease of solar radiation, turbulent kinetic energy and thereby suppression of the mixing layer. The feedback mechanism can explain the rapid formation of intense haze episodes to some extent, and we suggest that the detailed feedback mechanism warrant further investigation both from model simulations and field observations.

**1. Introduction**

With the rapid economic growth and urbanization, an increasing frequency of haze episodes along with the air pollution has become of great concern in China during the last decade (Cao et al., 2016; Huang et al., 2014; Kulmala, 2015; Wang et al., 2014; Wang et al., 2015). For example, during December 2016 a series of intense haze episodes took place in Eastern China, characterized by surface $PM_{2.5}$ concentrations exceeding 500 ug m$^{-3}$ in several measurement sites in Beijing and its surrounding sites (http://www.mep.gov.cn/gkml/hbb/qt/201701/t20170102_393745.htm). Severe air pollution has serious effects on human health. A recent study reported that the particulate matter has significantly decreased the life span of residents as many as 5.5 years in Northern China ( Chen et al, 2013). In a global scale, the air pollution was estimated to cause over 3 million premature deaths every year (Lelieveld et al., 2015).

Increased emissions from fossil fuel combustion due to vehicle traffic, industrial activities and power generation, along with exceptionally strong secondary aerosol formation, were thought to be responsible for these haze episodes (Cheng et al., 2016; Huang et al., 2014; Pan et al., 2016; Petäjä et al., 2016; G Wang et al., 2016a; Zhang et al., 2015; Zhao et al., 2013). Meanwhile, the formation of intense haze episodes was considered to be affected by meteorological conditions (Wang et al.,

2014; Quan et al., 2013; Wang et al., 2016b; Zheng et al., 2016). For example, the mixing layer
height is a key parameter that constrains the dilution of surface air pollution, and the development
of mixing layer is highly related to the amount of solar radiation absorbed by the air and reaching
the surface (Ding et al., 2016; Stull, 1988; Sun et al., 2013; Tang et al., 2016; Wilcox et al., 2016).
By using filed measurements combined with model simulation, a positive feedback between aerosol
pollution, relative humidity and boundary layer was found to be important in aerosol production,
accumulation and severe haze formation in Beijing (Liu et al., 2018).  Wang et al. (2018) found that
PBL schemes in their atmospheric chemistry models are not sufficient to describe the explosive
growth of PM2.5 concentration in Beijing-Tianjin-Hebei region due to absence of an online
calculation of aerosol-radiation feedback, and/or a deficient description of extremely weak turbulent
diffusion.

In this study, using unique measurements on the Beijing 325-meter-high meteorology tower, we
show clear relationship between mixing layer height and turbulent kinetic energy at the 140-m
observation platform. We also present direct evidence on the feedback that relates the decreasing
mixed layer height with increasing particulate matter concentrations, and this feedback is critical to
the formation of intense haze episodes in Beijing.
**2.   Methods**

84        2.1  Calculation of mixing layer height with ceilometer

85        The ceilometer was deployed in the yard of IAP (Institute of atmospheric physics, Chinese

academy of science), with a horizontal distance around tens of meters from the 325-m meteorology
tower. The mixing layer height was measured with the enhanced single-lens ceilometers from July
of 2009 to August of 2012 (CL 31, Vaisala, Finland), which utilized the strobe laser lidar technique
(910 nm) to measure the attenuated backscattering coefficient profiles. Detection range of the CL31
is 7.6 km with the report period of 2-120 s. Detail information can be found in previous studies
(Tang et al., 2016). Since the distribution of particle concentrations is uniform in the mixing layer
and has significant differences between the mixing layer and free atmosphere, the height at where a
sudden change exists in the attenuated backscattering coefficient profile indicates the top of the
mixing layer height. The Vaisala software product BL-VIEW was used to determine the mixing
layer height by finding the position with the maximum negative gradient ($-d\beta/dx$) in the attenuated
backscattering coefficient profiles as the top of the mixing layer (Münkel et al., 2007).

2.2 Measurements of energy flux at the 325-m Beijing meteorology tower
The turbulent fluxes of sensible heat ($Q_H$), latent heat ($Q_E$) and the turbulence kinetic energy (TKE)
were measured at the 140-m level using eddy covariance technique from July of 2009 to August of
2012. The raw data (10 Hz) of wind components (u, v and w) and sonic temperature (Ts) recorded
with three-dimensional sonic anemometers (Model CSAT3, Campbell Scientific Inc., Logan, Utah,
USA) and of water vapor concentrations (q) with open-path infrared gas analyzers (Model LI-7500,
LiCor Inc., Lincoln, Nebraska, USA). The fluxes of heat (Q) were calculated as the covariance
between the instantaneous deviation or fluctuations of vertical velocity ( $w_i'$ ) and their respective
scalar ( $s_i'$ ) averaged over a time interval of 30 min:

$$Q = \overline{w's'} = \frac{1}{N}\sum_{i=1}^{N} w's'$$

Where the over-bar denotes a time average, N is the number of samples during the averaging time
and the fluctuations are the differences between the instantaneous readings and their respective
means. The TKE were calculated as follows (stull,1988):

$$\frac{TKE}{m} = \frac{1}{2}\left(\overline{u'} + \overline{v'} + \overline{w'}\right) = \overline{e}$$

where m is the mass (kg), e is the TKE per unit mass ($m^2$ $s^{-1}$). A more detailed description of the
calculation and post processing of flux is provided elsewhere (Song et al., 2013).

2.3 Measurements of $PM_{2.5}$ concentration and gases at the 325-m Beijing meteorology tower.
The mass concentration of $PM_{2.5}$ at 8-m, 120-m and 280-m observation platforms were measured
with three TEOM RP1400 simultaneously from July of 2009 to August of 2012. (Thermo Scientific,
http://www.thermoscientific.com). The resolution and precision of the instrument for one-hour
measurements were 0.1 µg $m^{-3}$ and ±1.5 µg $m^{-3}$, respectively. The filters were exchanged when the
loading rates were approximately 40%. The flow rate was monitored and calibrated monthly. The
volume mixing ratios of ozone and NOx were measured with 49i and 42i (Thermal Environment
Instruments (TEI) Inc.), respectively (Wang et al., 2014).

2.4 Experiment design
The model used in this study is the Weather Research and Forecasting (WRF) model (ARW, version
3.8.1; Skamarock et al. 2008). The simulation domain was centered in Beijing (39.0°N, 116.0°E)
and implemented with one-nested grids with a resolutions of 1 km. The number of grid cells was
$460 \times 403$ for the domain in the east-west and south-north directions. The model run was initialized
at 00:00 UTC (or 08:00 LST) 16 Nov 2010 and integrated for 131 h until 10:00 UTC 21 Nov 2010,
including 48 h for spin-up. The initial conditions of the model and its outermost lateral boundary
conditions, as well as the soil moisture field, were taken from National Centers for Environmental
Prediction/National Center for Atmospheric Research Reanalysis data (resolution: $1° \times 1°$). The
model physics schemes used include: Thompson microphysical parameterization (Thompson et al.,
2004); BouLac boundary-layer parameterization (Bougeault and Lacarrere 1989); RRTMG (Iacono
et al., 2008) radiation Scheme; The Building Effect Parameterization (BEP) and the Building Energy
Model (BEM) schemes implemented in WRF that can more accurately describe three-dimensional
urban land surface features and processes, including anthropogenic heat from buildings (Martilli et
al., 2002; Salamanca and Martilli, 2010). The control and test experiment were performed separately
to investigate impact of aerosol direct radiative forcing on surface temperature, relative humidity
and development of boundary layer height.  The control run (CTL) used the RRTMG radiation
scheme which ignored the direct radiation effects of aerosols input. In sensitivity test experiment,
we add the aerosol input in RRTMG scheme using Tegen climatology and urban type aerosols
during the sensitive test.

2.5 Other supporting measurements
Total solar radiation was measured with a direct radiometer (TBQ-2, Junzhou, China). Direct
radiation was measured with a direct radiometer (TBS-2, Junzhou, China). UV radiation in the range
of 220-400 nm was measured using CUV3 radiometer (USA). The estimated experiment error for
the three instruments are 3%, 1% and 2%, respectively. The original data were obtained at one-
minute intervals and the hourly average values were used in this study. The chemical composition
of organic, sulphate, nitrate, ammonium and chloride in non-refractory submicron aerosol were
measured during several campaigns with an Aerodyne High-Resolution Time-of-Flight Aerosol
Mass Spectrometer from July of 2009 to August of 2012 (HR-ToF-AMS, Aerodyne Research Inc.,
Billerica, MA, USA). Detailed information about instrument, calibration and data process have been
introduced by. All these measurements were conducted in the IAP station.

**3    Results and Discussion**
A typical intense haze episode occurred during the heating season in urban Beijing during 17 to 22
November 2010. This episode was associated with synoptic stagnation in the North China Plain
(Figure S1) and was characterized by low wind speeds and irregular wind direction (Figure 1).
Several meteorological variables had distinct temporal patterns during different stages of pollution,
including reduced solar radiation and increased relative humidity during the most intense presence
of haze (Fig. 1). The temporal patterns of $PM_{2.5}$ concentrations were very similar at the two lower
measurements heights (8 m and 120 m, Fig. 1d), even though the concentration was clearly the
highest close to the surface. The $PM_{2.5}$ concentration measured at 280 m behaved in a different way,
especially during the most intense period of the haze when the mixed layer height was very low
(Fig. 1e). The decoupling of the 280-m platform from the other two lower ones at low mixed layer
heights is apparent in our 3-year measurement data set, especially when comparing $O_3$ and $NO_x$
concentrations between the three measurement platforms (Figs. S2 and S5). During the haze period,
the maximum $PM_{2.5}$ concentrations at 8, 120 and 280 m were 505, 267 and 339 μg m$^{-3}$, respectively.
The higher maximum concentration at 280 m compared with 120 m can be ascribed to the transport
of pollutants from surrounding regions of Hebei and Tianjin Provinces typical for polluted periods
(Sun et al., 2013). The mixing layer height varied from 130 m to 1640 m during the haze episode,
ranging between about 200 and 500 m during the most intense period of the haze period on 18
November 2010 (Fig. 1e). The TKE was quite low during this intensive haze episode from 18
November to 21 November, with an average value around 0.3 m$^2$ s$^{-2}$. However, the TKE increased
significant on morning of 21 November as surface wind increased from 1.2 m/s to around 6 m/s,
which was possible due to the movement of cold front as shown in Figure S1.

The vertical distribution of attenuated backscatter density obtained from ceilometer measurements
indicate vertical mixing conditions accompanied with an inversion layer and high relative humidity
in the surface as shown in Figure 2. The strong inversion and high relative humidity occurred on
morning of 18 November 2010, with a lapse rate of 2K / 100 m, relative humidity of 78% and north-
direction wind speed of around 2 m / s detected by the vertical sounding. The turbulent kinetic
energy at 140 m was reduced to around 0.1~0.7 m$^2$/s$^2$ due to decreased solar radiation, as presented
in Figure1(a). In this manner, the development of a mixing layer was significantly suppressed during
the intense haze episode.

In order to demonstrate how the $PM_{2.5}$ modifies the surface temperature, relative humidity and
development of the mixing layer height. we performed two numerical simulation experiments, using
the WRF model as a tool. We took the measurements during the intensive haze episode shown in
Figure 1 as an example. As shown in Figure 3(a), the variation of temperature and relative humidity
showed pronounced daily variations, with higher and lower values, respectively, during daytime in
both test and control experiment. However, the presence of aerosol in the test experiment clearly
showed decreased surface temperature and increased relative humidity. The presence of aerosol
reduces downward radiation reaching the surface, as a result of which the surface temperature and
sensitive heat flux decrease, and the development of mixing layer height is suppressed (Li et al.,
2017a, 2017b; Miao et al., 2016). Statistical results showed that the average relative humidity,
surface temperature and mixing layer height were 8.2±3.4 °C, 40.5±11.6% and 377.7±499 m,
respectively, without the consideration of aerosol direct radiative forcing, whereas the consideration
of aerosol directive radiative forcing changed these values to 7.1±3.1 °C, 40.6±11.7 % and
326.7±470.1 m, respectively. Our model results clearly demonstrate the pronounced role of aerosol
particles in reducing the mixing layer height during this haze pollution episode.

In order to further illustrate how the mixing layer height modifies $PM_{2.5}$ concentrations, we used
three years of simultaneous winter-time air pollutant measurements in the Beijing. We divided the
observed $PM_{2.5}$ concentrations into highly-polluted and less-polluted conditions using a threshold
value of 75 μg m$^{-3}$ for $PM_{2.5}$ to distinguish between these conditions. This is consistent with
Chinese Environment Protection Bureau definition of a haze pollution events. With this threshold
value, we found that 31% and 69% of total measurement time corresponded to highly-polluted and
less-polluted conditions, respectively. We plotted the $PM_{2.5}$ data as a function of the mixing layer
height at the three observation heights (8 m, 120 m and 280 m) during both highly-polluted and
less-polluted conditions and fitted an exponential curve to these data based on best fitting (Figure.
4). The $PM_{2.5}$ concentration has a clear anti-correlation with the mixing layer height during the
intense haze episodes. At all the measurement heights, the $PM_{2.5}$ concentration increased as the
mixing layer height decreased, and this pattern was very strong under polluted conditions (Figure.
4). We also tested the reciprocal fitting function for the data (Figure S8). It overestimated the
$PM_{2.5}$ concentration when the mixing layer height was very low, as compared to the exponential
fitting function (Figure. 4). This also indicates that a much higher $PM_{2.5}$ concentration is needed in
order to obtain a very low mixing layer height without the positive feedback. This can also be
supported by the root-mean-square error (RMSE) of these two fitting methods. The RMSE of the
exponential fitting is much smaller than the reciprocal fitting in any case (Table. S1).

It is worth noting that the increase was mainly from the $PM_{1-2.5}$ fraction that increased from 42% to
65% as mixing layer height decreased from more than 1400 m to lower than 300 m (Figure S4). A

major portion of particulate mass between 1 and 2.5 μm originates from secondary aerosol formation processes in urban air (Wang et al., 2014; Zhang et al., 2015). As shown in Figure S7, the concentration of NR-PM$_1$ increased significantly from 12.1 μg m$^{-3}$ to 56.4 μg m$^{-3}$ with the variation of MLH decreased from more than 1400 m to less than 200 m. The reduction in solar radiation reaching the surface due to fine particle matter reduces the turbulent kinetic energy and the development of mixing layer, as shown in Figure.5. An exponential function between the turbulent kinetic energy at 140 m and mixing layer height was fitted., Based on this fit, the MLH roughly be doubles from about 400 m to 800 m when TKE increases from 0.1 m$^2$ s$^{-2}$ to 1 m$^2$ s$^{-2}$. These are typical values of MLH during polluted conditions in Beijing.

The reduced sensible heat and TKE due to aerosol particles reduces the entrainment of relatively dry air into mixing layer from above, which makes the air more humid within the mixing layer. This, together with the decreased surface temperature increases the relative humidity (Li et al., 2017b). The increased relative humidity enhances the aerosol water uptake and promotes the formation of secondary organic and inorganic aerosol via aqueous phase reactions (Liu et al., 2018; Wang et al., 2019), .enhancing light scattering and causing further reduction in the intensity of radiation reaching the surface. All these factors suppress the development of mixing layer height and enhance the accumulation of air pollutants within the mixing layer. We ascribe part of the observed increase in PM$_{2.5}$ and simultaneous decrease in the mixing layer height to the positive feedback associated with the particulate matter-mixing layer interaction (Petäjä et al. 2016, Ding et al. 2016), occurring at the same time as primary emissions and secondary formation are confined into a smaller volume of air. The feedback occurred at all the three observation platforms and appeared to be most intensive at 8 m. In an urban environment, NO$_x$ originates mainly from local anthropogenic emissions, whereas the sources of particulate matter include both primary emissions and secondary formation (Ehn et al., 2014; Jimenez et al., 2009; Zhang et al., 2015; Zhao et al., 2013). As shown in Figure S6, the median NO$_x$ concentration at 8 m was 250% higher under highly polluted conditions compared with less-polluted conditions as the mixing layer height decreased to 100-200 m, while the corresponding number for the PM$_{2.5}$ concentration was 360%.

The increase of the PM$_{2.5}$ concentration from less-polluted to highly-polluted conditions is mainly due to concentrated particulate matter caused by a decreased mixing layer height, which is accompanied by primary particle emissions, secondary aerosol formation and feedback from particulate matter-mixing layer height interactions. Compared with the increased amounts of NOx, we can roughly estimate that in maximum 110% of the increased PM$_{2.5}$ originates from secondary aerosol formation processes in this study. Of the remaining 250% of the PM$_{2.5}$ increase, potentially

a large fraction originates from particulate matter-mixing layer height interactions, but we cannot
quantify this fraction at the moment.

4 **Conclusions**
The development of mixing layer height in an urban city is affected by the intensity of
incoming solar radiation. Our measurement at the 325-meter meteorology tower showed that the
solar and ultraviolet radiation reaching the surface decrease considerably at increased pollution
levels, which leads to a decreased TKE and, consequently, the suppression of mixing layer
development. In turn, the shallowed mixing layer height further favors the enhancement of $PM_{2.5}$
concentration and its precursor gases from both direct emissions and secondary formation. This
feedback mechanism may be an important reason for rapid increase of particulate matter from
moderate-polluted conditions to periods of intense pollution in an urban atmosphere as the strength
increased with the $PM_{2.5}$ concentration increased, although we cannot quantify the feedback amount
exactly by observations currently. The particulate matter-mixing layer height feedback is probably
a critical factor for the formation of intense haze periods from moderate-polluted periods in Beijing
and other polluted cities.
**Acknowledgements**
This work was supported by the Ministry of Science and Technology of China (No:
2017YFC0210000), the Ministry of Science and Technology of China (No: 2017YFC0210102), the
National Research Program for key issues in air pollution control(DQGG0101), Beijing National
Science Foundation of China (8171002) and Academy of Finland via Center of Excellence in
Atmospheric Sciences.
**Competing financial interests**
The authors declare no competing financial interests.
**Author contributions**
M.K, T.P and Y.H.W, have the original idea of the research. Y.S.W, G.T, T.S, Z.L, B.H, L.W, X.Z,
D.J, W.G and Y.S conducted the longtime measurements and provided the data. M.Y conducted
model simulation. Y.H.W, G.T, S.T, P.Z, M.E, C.Y, V.K, T.P and M.K interpreted the data and
plotted the figures. Y.H.W wrote the manuscript, with contribution from all co-authors.

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

Figure captions

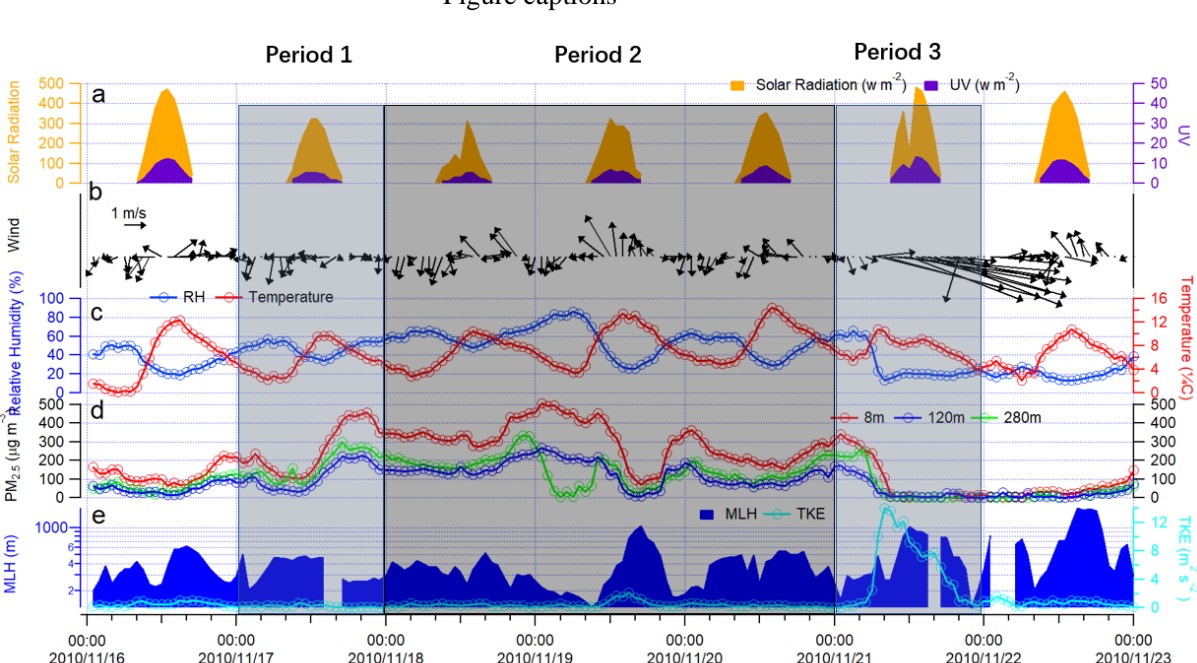




**Figure 1**. Measurements of (a) solar radiation and ultraviolet radiation at 8 m, (b) wind speed and
direction at 8 m, (c) relative humidity and air temperature at 8 m, (d) mass concentration of $PM_{2.5}$
at 8 m, 120 m and 280 m, (e) mixing layer height at 8 m and turbulence kinetic energy at 140 m in
the Beijing 325-meter meteorology tower during an intensive air pollution episode in November of
2010. The evolution of the air pollution episode can be divided into the period 1 (clean period to
air pollution accumulation period, period 2 (pollution period) and period 3 (pollution to clean
period).

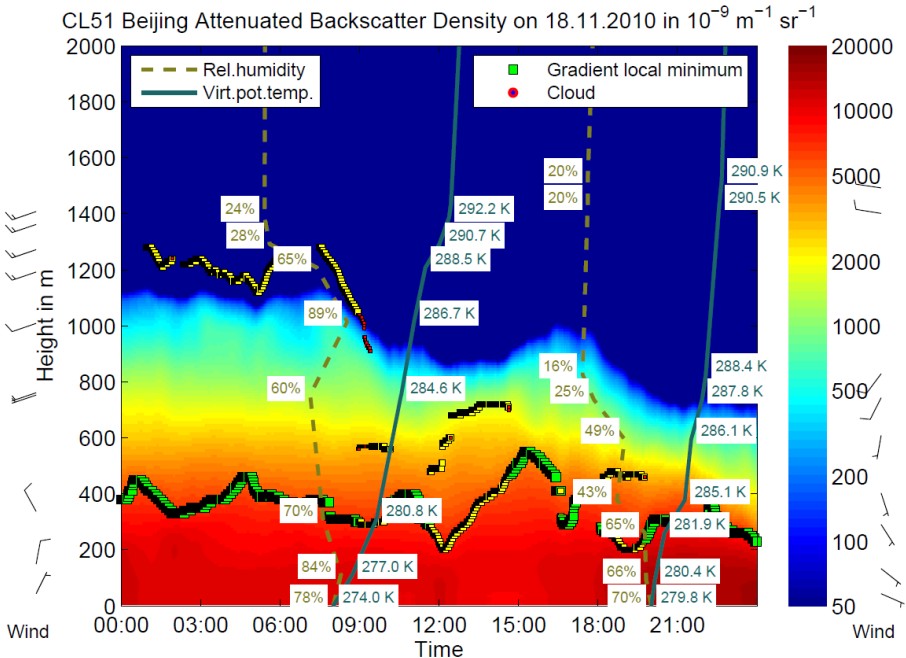


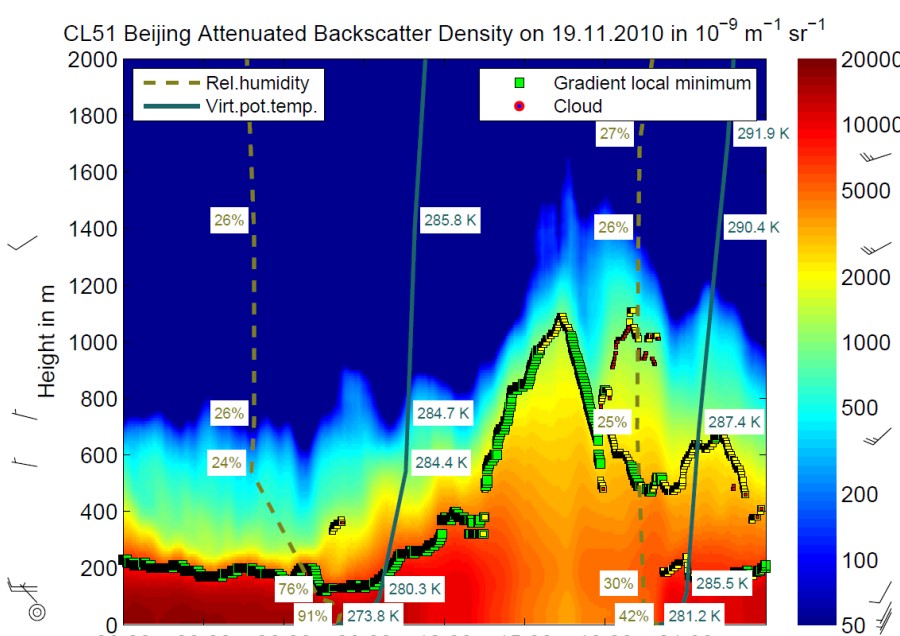


**Figure 2**. Observed attenuated backscatter density, calculated mixing layer height using ceilometer and vertical wind speed, wind direction, relative humidity, virtual potential temperature using sounding data during November 18 (top) and 19 (bottom). The black flag in the left and right side of the figures stand for vertical wind speed and wind direction obtained from sounding measurements at 08:00 and 20:00 of Beijing time, respectively. The circle in the left side of figure represents calm wind. The dotted yellow lines and solid green lines represents vertical distribution of virtual potential temperature and relative humidity from sounding at 08:00 and 20:00,

respectively. The yellow square and green square represent first layer and second layer, respectively,
and usually the first layer was used as mixing layer height. The mixing layer height was determined
from the local minimum of the backscatter density gradient, and the colour in the figure stands for
backscatter density from ceilometer. From both figures, we can clearly see that mixing layer has
important role in regulating distribution of air pollutants.


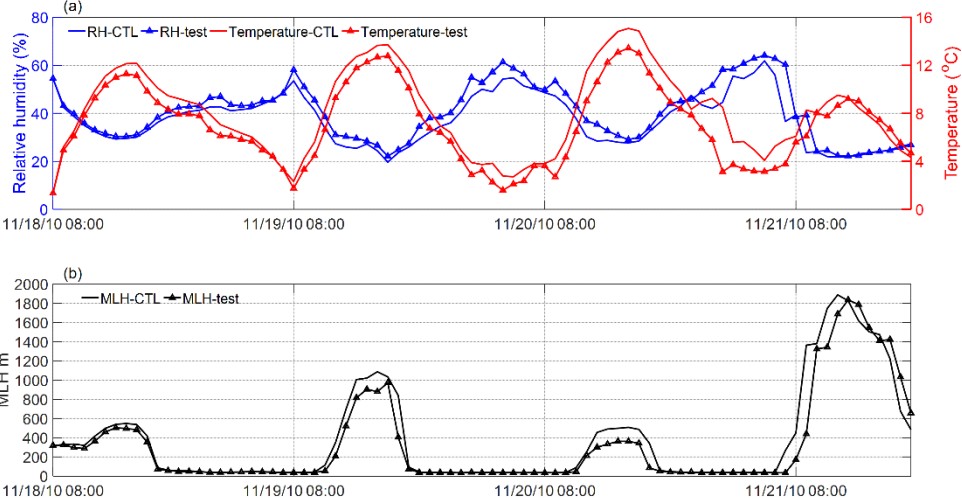


Figure 3(a) Modeled variation of surface relative humidity, temperature and (b) mixing layer height during the intensive haze episode from 18th November 2010 to 21th November 2010. The lines with triangles on represent results from test experiment, while the lines represent results from control experiment. The control experiment was performed with absence of aerosol direct radiative forcing in the RRTMG radiation scheme, while the test experiment was conducted with presence of aerosol direct radiative forcing considered.

















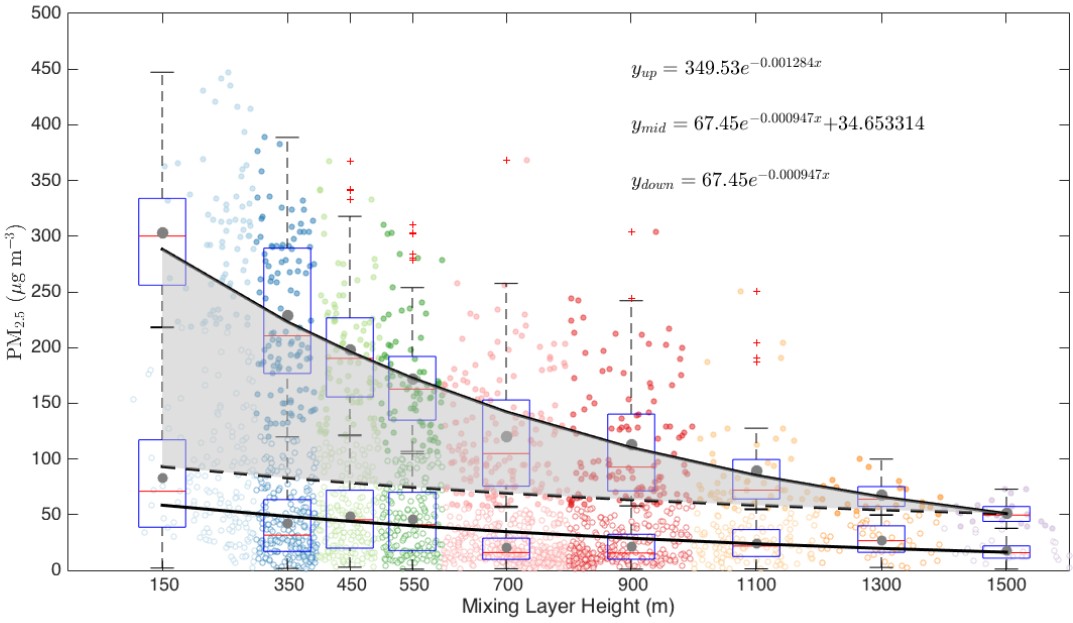



(a)

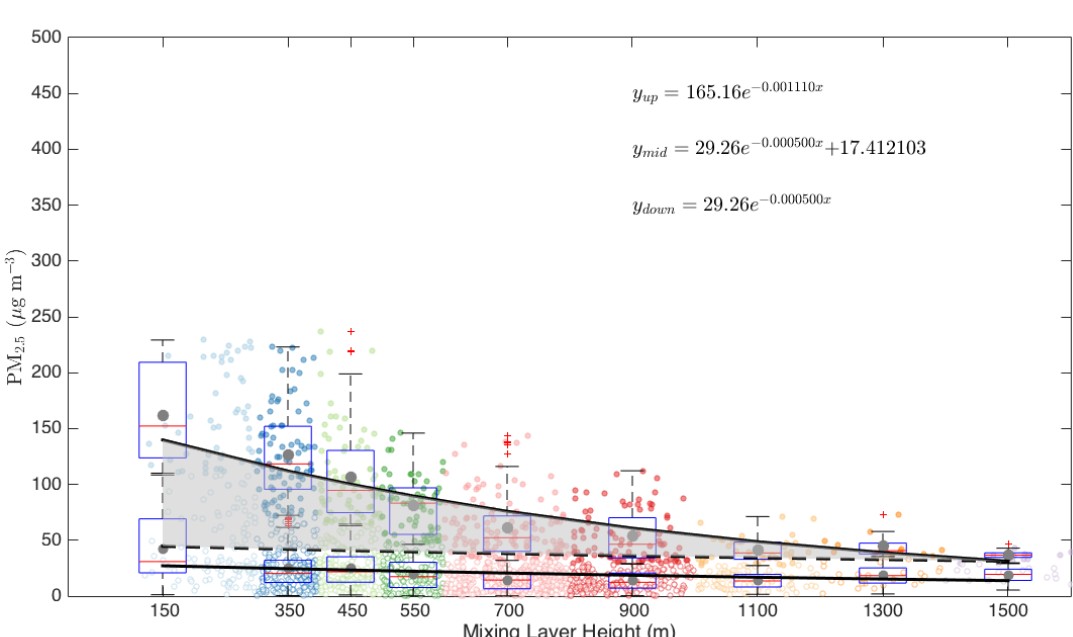



(b)

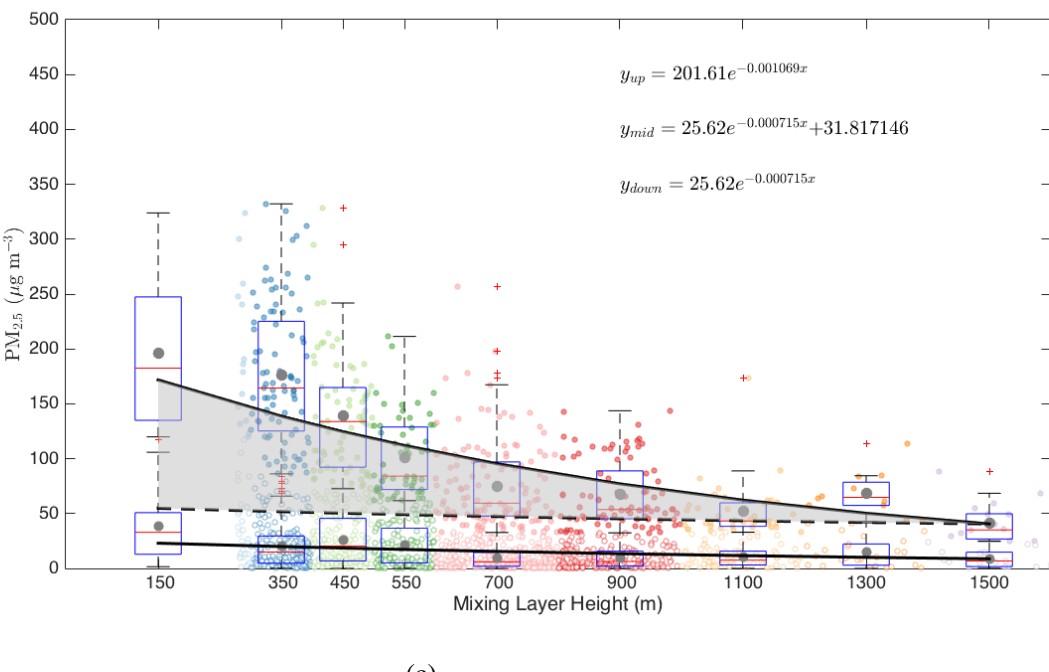

$$y_{up} = 201.61e^{-0.001069x}$$

$$y_{mid} = 25.62e^{-0.000715x}+31.817146$$

$$y_{down} = 25.62e^{-0.000715x}$$

(c)

**Figure 4.** The variability of the PM$_{2.5}$ mass concentration as a function of the mixing layer height at 8 m (a), 120 m (b) and 280 m (c). The data related to the upper fitting line represents PM$_{2.5}$ concentrations larger than 75 ug m$^{-3}$, while the data related to the lower fitting line represents PM$_{2.5}$ concentrations less than 75 ug m$^{-3}$. The dark grey points represent mean values; the red line represents median values. The shadowed area corresponds to an increased amount of PM$_{2.5}$ with decreased mixing layer height assuming that PM$_{2.5}$ has the same variation pattern under highly- polluted conditions as in less polluted time.












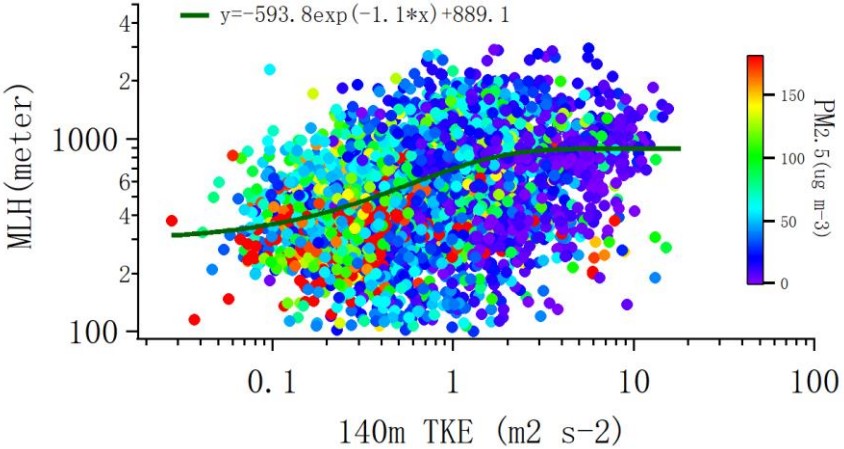


**Figure 5.** Turbulent kinetic energy at 140 m as a function of mixing layer height and PM$_{2.5}$ concentrations at 120 m from July of 2009 to August of 2011. An exponential function was fitted based on best fitting.
