# Peer review of "Rapid formation of intense haze episodes via aerosol-boundary layer feedback in Beijing"

_Atmospheric Chemistry and Physics, 2018_

## Referee Comment (RC1) · Anonymous Referee #1 · 24 Nov 2018

This paper presents valuable observational evidence for interactions between boundary layer mixing properties and haze aerosol loading. It describes an important set of long-term measurements from a polluted urban location in Beijing, combining atmospheric composition measurements at a number of different altitudes with ceilometer measurements of atmospheric mixing height. This provides a unique opportunity to provide conclusive evidence for a feedback between aerosol and mixing height, but unfortunately this has not been fully achieved, and the claims that are made are not fully supported.

The study shows that PM is higher when the mixing height is lower, but does not present deeper analysis. The inverse relationship between aerosol and mixing height may represent a dynamical feedback, or it may just reflect the one-way control of PM

levels by mixing height. It is ascribed to the former here (line 178), but the evidence provided to back this up is weak, and no argument is presented on why the simpler mechanism (reduced vertical mixing alone) is insufficient to explain it. This needs to be addressed before the paper is suitable for publication.

The ratio between NOx and PM shown in Fig S5 could potentially provide valuable new insight into the mechanisms involved, but this is not explored in the paper. Similarly, increases in secondary aerosol production shown in Fig S6 are interesting but not properly explored. Neither Fig S5 nor Fig S6 are even referred to in the paper.

The paper calls for the feedback between aerosol and boundary layer turbulence to be included in air quality models. In practice online models already include this process, and what is missing is strong observational evidence to back up existing theory and validate our current understanding. The study described in this paper has the potential to provide this important evidence, and I strongly encourage the authors to complete this analysis and revise their paper, after which it could make a valuable and important contribution to the scientific literature.

General Comments

The conclusions of the paper remain weak and speculative, principally because clear evidence has not been provided in the paper to support the suggestions made.

The figures in the supplementary material are not labelled or described well, and their relevance to the body of the paper is unclear. Please justify their inclusion carefully. Figures S5-S7 are not referred to in the text, and should be introduced or removed. Further specific comments are provided below.

The written English in the paper is reasonable but does not fully reach the standards needed for a scientific publication, and some polishing of this will be required in the final version.

Specific Comments

Page 3: particle concentrations are often not uniform in the mixed layer, particularly near sources or where there are residual layers present. How does the approach used resolve multiple layering? The maximum gradient approach used has known deficiencies; how sensitive is the mixing height retrieval to the approach taken?

Line 80: Please describe where these measurements were made.

Line 132: Fig 1 does not show reduced temperatures: the temperature appears higher than before or after

Line 154-156: The study of TKE is essential for supporting the links between mixing height and PM levels, but it is not mentioned again except in reference to Fig 4 in the following paragraph.

Line 167: What is the justification for fitting an exponential curve to the data in Figure 3? Fitting a reciprocal curve would make more sense (doubling mixing height halves concentrations). Please explain the rationale here and the evidence supporting it.

Line 171-7: These arguments are the most important made in the paper, and need clear observational support. What relationship is shown in Figure 4? What is the signal underlying the scatter? Extracting a quantitative relationship from the data shown here and using it to support the arguments made would strengthen the paper greatly.

Line 202: Anthropogenic heating isn't mentioned in the paper prior to this.

Line 316: The references from this point onward are out of alphabetic sequence in the reference list.

Line 391: What is the distinction between the yellow and green mixing height lines in Figure 2? This needs to be explained more clearly in the caption.

Fig S1: Please simplify this figure (perhaps by reducing to 4 panels) as there is too much extraneous detail here. Annotate to highlight key features that you want the reader to be aware of. Please label High and Low pressure systems more clearly, and

label each panel with the date and time.

Fig S2: Make figure clearer, label panels and state the time period used. There is no analysis of this figure in the text; please state how it contributes to the arguments in the paper (or otherwise remove it).

Fig S3: The panels in this figure are small and difficult to read, and there is no apparent connection between them. If they are needed, present as two separate figures. In the first, explain what is meant by direct radiation (what wavelength range? excluding all diffuse components?) and use a compatible color scale between the two panels so that the results can be compared easily (dark blue is 8m haze in one panel and 280m clean in the other). X-axis labels are missing in the PM1 figure.

Fig S4: Please state the time period and explain x-axis (no axis label is provided)

Fig S5: Define Fb here. This figure does not appear relevant to the paper, and should be removed.

Fig S5(2): Please correct the numbering of this and subsequent figures

Fig S6: The increase in secondary aerosol is interesting here, but it would be more informative to show how the contribution of secondary aerosol to total aerosol changes here. However, the figure is not discussed in the paper, and should either be discussed or removed.

Fig S7: What period is this over? Again, this figure should be discussed or dropped.

Typos and minor issues

Please use European date order conventions (day, month, year) throughout. The x-axis labels in Fig 1 use US conventions, while Fig S5 in the supplement uses Chinese conventions.

Title: episode -> episodes

line 107: in -> on (or at)

line 111: remove to

line 118: 220-400?

line 142: microgram unit missing

line 178: assign -> ascribe

line 184: bot ??

line 396-7: stand for -> represents

line 415: date -> data

---

## Referee Comment (RC2) · Anonymous Referee #2 · 2 Jan 2019

This paper characterizes the interactions between atmospheric mixing layer dynamics and fine particulate matter pollution using long-term measurements of vertical distributions of PM2.5 and NOx, atmospheric mixing layer height, vertical meteorological parameters, energy flux, etc. in an urban site in Beijing. Based on the relationship between PM2.5 concentration, mixing layer height, solar radiation, and turbulent kinetic energy, the authors claimed that they found a feedback mechanism between mixing layer height and fine particulate matter pollution that could explain the rapid formation of severe haze pollution episodes in Beijing.

This work addresses an important topic that are of interest to many of the readers in atmospheric science community. However, many data presented in the paper are not thoroughly analyzed and discussed, and the evidence claimed by the authors are

not strong enough to support their conclusion about the aerosol pollution-mixing layer development feedback mechanism. There are many important issues that need to be addressed before the publication of the paper in ACP can be considered.

Major comments:

The authors claimed that the fine aerosols can reduce the solar radiation reaching the surface, resulting in a decrease in the turbulent kinetic energy (TKE) and a suppression of atmospheric mixing layer development, which further increase aerosol concentrations from direct emission and secondary formation (i.e., the feedback mechanism). However, they did not provide clear evidence that fine aerosols play a non-negligible role in regulating TKE and mixing layer heights. As shown in Fig 1, the TKE decreases dramatically from 8:00 to 20:00 on 21 November 2010, while aerosol loadings are pretty low during this period. This suggests that the variation of TKE is largely driven by non-haze related factors. Therefore, to claim the feedback mechanism, it is important to quantify to what extent fine aerosols can reduce or regulate the TKE and the development of the mixing layer in severe haze episodes.

Other specific comments:

In the Introduction, the review of literatures is too brief. A summary of the current knowledge and remain issues regarding the interactions between boundary layer dynamics and aerosol pollution should be included, and the novelty of the present study should be clearly pointed out.

P4, Sect. 2.3 and 2.4. Please specify the altitude at which the measurements of O3, NOx, radiation, and aerosol chemical composition were performed.

Line 111. Remove "to", and "ratio" should be "rate".

Line 121-123. The HR-ToF-AMS was used to measure aerosol chemical composition. However, the data were not discussed in the paper, though a figure (Fig. S6) was included in the supplementary martial.

Line 138-141 and Figs. S2 and S4. The decoupling of the 280-m platform from the other two lower ones for O3 was shown to be much smaller than that for PM2.5 and NOx. What is the reason for this difference?

Line 171-173 and Fig. S3b. Was the PM1 measured by HR-ToF-AMS? If so, the authors should point out that measured PM1 mass concentrations do not include the refractory components such as soot and dust, whereas the PM2.5 concentrations include these components. In addition, compared to PM1-2.5, the origin of PM1 is generally more secondary. Therefore, the increase of PM1 concentration but decrease of its mass fraction in PM2.5 (as the decrease of mixing layer height) may offer insights into the contributions of primary emission and secondary formation to the haze pollution. This merits further discussions in the paper.

Line 184. "bot" should be "both".

Some references in the reference list do not follow an alphabetical order.

––––––––––––––––––––––––––––

---

## Author Comment (AC1) · 18 Mar 2019

The comment was uploaded in the form of a supplement:
https://www.atmos-chem-phys-discuss.net/acp-2018-1079/acp-2018-1079-AC1-supplement.pdf

---

## Author Comment (AC2) · 18 Mar 2019

<h1 style="text-align:center">A point to point response to the reviewers' comments</h1>

We thank the two reviewers for their comments, and we do think their comments and suggestions improved our manuscript a lot. Here are points to points responses (in blue colored), accordingly, we also revised manuscript (in red colored).

Reviewer #1

This paper presents valuable observational evidence for interactions between boundary layer mixing properties and haze aerosol loading. It describes an important set of long-term measurements from a polluted urban location in Beijing, combining atmospheric composition measurements at a number of different altitudes with ceilometer measurements of atmospheric mixing height. This provides a unique opportunity to provide conclusive evidence for a feedback between aerosol and mixing height, but unfortunately this has not been fully achieved, and the claims that are made are not fully supported.

The study shows that PM is higher when the mixing height is lower, but does not present deeper analysis. The inverse relationship between aerosol and mixing height may represent a dynamical feedback, or it may just reflect the one-way control of PM levels by mixing height. It is ascribed to the former here (line 178), but the evidence provided to back this up is weak, and no argument is presented on why the simpler mechanism (reduced vertical mixing alone) is insufficient to explain it. This needs to be addressed before the paper is suitable for publication.

The ratio between NOx and PM shown in Fig S5 could potentially provide valuable new insight into the mechanisms involved, but this is not explored in the paper. Similarly, increases in secondary aerosol production shown in Fig S6 are interesting but not properly explored. Neither Fig S5 nor Fig S6 are even referred to in the paper.

The paper calls for the feedback between aerosol and boundary layer turbulence to be included in air quality models. In practice online models already include this process, and what is missing is strong observational evidence to back up existing theory and validate our current understanding. The study described in this paper has the potential to provide this important evidence, and I strongly encourage the authors to complete this analysis and revise their paper, after which it could make a valuable and important contribution to the scientific literature.

Response: we thank the reviewer for careful reading and the comments. We think these comments are really important to improving the manuscript. We calcified the reviewer's comments to two comments, as following:1. The study shows that PM is higher when the mixing height is lower, but does not present deeper analysis. The inverse relationship between aerosol and mixing height may represent a dynamical feedback, or it may just reflect the one-way control of PM levels by mixing height. It is ascribed to the former here (line 178), but the evidence provided to back this up is weak, and no argument is presented on why the simpler mechanism (reduced vertical mixing alone) is insufficient to explain it. This needs to be addressed before the paper is suitable for publication. We agree that inverse relationship

between aerosol and MLH represent a dynamical feedback. The increase of PM will decrease MLH by reduce solar radiation and thereby TKE, and decreased MLH will also lead to more secondary formation due to concentrated precursors. Also, we carefully ascribed partly the increase in $PM_{2.5}$ and simultaneous decrease in the mixing layer height to a positive feedback from particulate matter-mixing layer interaction, since other factors may work during this process. For example, a recent study by Liu et al., (2018) suggested that relative humidity played an important role in feedback between secondary particle matter and boundary layer height. 2.The ratio between NOx and PM shown in Fig S5 could potentially provide valuable new insight into the mechanisms involved, but this is not explored in the paper. Similarly, increases in secondary aerosol production shown in Fig S6 are interesting but not properly explored. Neither Fig S5 nor Fig S6 are even referred to in the paper.

The sources of NOx and PM are mainly form surface emission and formation, therefore, the concentrations are both higher at lower observation platform than concentrations at higher observations platform. We added statements in our revised manuscript about figure s6, as following: As shown in figure S7, the chemical concentration of NR-$PM_1$ increased significantly from 12.1 μg m$^{-3}$ to 56.4 μg m$^{-3}$ with the variation of MLH decreased from more than 1400 m to less than 200 m

General Comments

The conclusions of the paper remain weak and speculative, principally because clear evidence has not been provided in the paper to support the suggestions made.
The figures in the supplementary material are not labelled or described well, and their relevance to the body of the paper is unclear. Please justify their inclusion carefully.
Figures S5-S7 are not referred to in the text, and should be introduced or removed.
Further specific comments are provided below.
The written English in the paper is reasonable but does not fully reach the standards needed for a scientific publication, and some polishing of this will be required in the final version.
Response: Thanks for the comment. We have revised the manuscript on grammars and writing.

Specific Comments

Page 3: particle concentrations are often not uniform in the mixed layer, particularly near sources or where there are residual layers present. How does the approach used resolve multiple layering? The maximum gradient approach used has known deficiencies; how sensitive is the mixing height retrieval to the approach taken?
Response: Thanks for the comment. The maximum gradient approach was used in our study to retrieve mixing layer height. Utilizing with approach with ceilometer measurements, three layers can be retrieved within 4000 m height from surface. Usually, the first layer was considered as mixing layer height. In our previous study, we found that mixing layer height retrieved by this method was consistent with sounding data results during polluted days, while the method can not be used during dust conditions and neutral boundary layer (Tang et al., 2016). As the reviewer has pointed out clearly that the distribution of particle matter is not uniform in the mixed layer, particularly during a stable boundary layer occurred. We can not get exact distributions of PM in the vertical scale, so the assumption of relative uniform of PM in the mixed layer could be carefully made. We also recognize that detailed technic work is challenge and needed in future.

Line 80: Please describe where these measurements were made.

Response: Thanks for the comment. The measurements were made in the yard of institute of atmospheric physics (IAP), where the 325 meter tower is located. We also added descriptions on our revised manuscript.

Line 132: Fig 1 does not show reduced temperatures: the temperature appears higher than before or after

Response: Thanks for the correction. We corrected this statement in our revised version.

Line 154-156: The study of TKE is essential for supporting the links between mixing height and PM levels, but it is not mentioned again except in reference to Fig 4 in the following paragraph.

Response: Thanks for the comment. We added more discussion about variation of TKE in Figure 1. The TKE was quite low during this intensive haze episode from 18 November to 21 November, with an average value around 0.3 $m^2s^{-2}$. However, the TKE increased significant on morning of 21 November as surface wind increased from 1.2 m/s to around 6 m/s, which was possible dur to cold front as shown in Figure S1.

Line 167: What is the justification for fitting an exponential curve to the data in Figure 3? Fitting a reciprocal curve would make more sense (doubling mixing height halves concentrations). Please explain the rationale here and the evidence supporting it.

Response: Thanks for the comment. In general, the reciprocal fitting can not represent a positive feedback. So in this case, the exponential fitting curve is more appropriate. This can also be supported by the root-mean-square error (RMSE) of these two fitting methods. The RMSE of the exponential fitting is much smaller than the reciprocal fitting in any case (Table R1).

Moreover, we have tested the reciprocal fitting function for the data as shown in Figs. R1, R2, and R3. It overestimated the PM 2.5 concentration when the mixing layer height is very low compared to the exponential fitting function (Fig. 3). This also indicates that a much higher PM 2.5 concentration is needed in order to obtain a very low mixing layer height without a positive feedback.

Table R1: The root-mean-square error (RMSE) in the unit of µg m$^{-3}$ between fitted functions and measured data.

| Fitting function | Pollution intensity | At 8 m | At 120 m | At 280 m |
|---|---|---|---|---|
| reciprocal | High | 53.5589 | 27.8437 | 69.7236 |
| reciprocal | Low | 12.0393 | 8.2159 | 9.3756 |
| exponential | High | 9.2853 | 11.4142 | 19.0527 |
| exponential | Low | 10.4951 | 6.2643 | 6.6181 |

8m

120 m

[Figure]

280m

Figure r1 . The variability of the $PM_{2.5}$ mass concentration as a reciprocal function of the mixing layer height at 8 m (R1), 120 m (R2) and 280 m (R3).

Line 171-7: These arguments are the most important made in the paper, and need clear observational support. What relationship is shown in Figure 4? What is the signal underlying the scatter? Extracting a quantitative relationship from the data shown here and using it to support the arguments made would strengthen the paper greatly.

Response: Thanks for the comment. We fitted an exponential function between Turbulent kinetic energy at 140 m and mixing layer height.   We also added some more statement in our

revised manuscript. An exponential function between Turbulent kinetic energy at 140 m and mixing layer height was fitted, which could provide us some simple quantification. As presented in the function, the MLH will be doubled from around 400 m to 800 m if TKE increased from around 0.1 $m^2\,s^{-2}$ to 1 $m^2\,s^{-2}$, and these are typical MLHs during polluted conditions in Beijing.

[Figure]

Line 202: Anthropogenic heating isn't mentioned in the paper prior to this.
Response: Thanks for the comment. We removed it in our revised version.
Line 316: The references from this point onward are out of alphabetic sequence in the reference list.
Response: Thanks for the comment. We made it in the corrected order.
Line 391: What is the distinction between the yellow and green mixing height lines in Figure 2? This needs to be explained more clearly in the caption.
Response: Thanks for the comment. The yellow square and green square stand for first layer and second layer, respectively, and usually the first layer was used as mixing layer height.
Fig S1: Please simplify this figure (perhaps by reducing to 4 panels) as there is too much extraneous detail here. Annotate to highlight key features that you want the reader to be aware of. Please label High and Low pressure systems more clearly, and label each panel with the date and time.
Response: Thanks for the comment. We reduced the figures to 4 panels and added date and time in each panels, explanations in figure captions part about the high and Low pressure systems.
Fig S2: Make figure clearer, label panels and state the time period used. There is no analysis of this figure in the text; please state how it contributes to the arguments in the paper (or otherwise remove it).
Response: Thanks for the comment. Actually, the air pollutants were well mixed during daytime, primary pollutants showed higher concentration near surface, while the secondary pollutant, ozone showed higher concentrations in higher altitude. We have analysis at Line 141. The decoupling of the 280-m platform from the other two lower ones at low mixed layer heights is apparent in our 3-year measurement data set, especially when comparing $O_3$ and $NO_x$ concentrations between the three measurement platforms (Figs. S2 and S5).
Fig S3: The panels in this figure are small and difficult to read, and there is no apparent connection between them. If they are needed, present as two separate figures. In the first, explain what is meant by direct radiation (what wavelength range? excluding all

diffuse components?) and use a compatible color scale between the two panels so that the results can be compared easily (dark blue is 8m haze in one panel and 280m clean in the other). X-axis labels are missing in the PM1 figure.

Response: Thanks for the comment. The direct radiation is the proportion of the almost rectilinear solar radiation, which reaches the earth's surface from an angle with a distance of 0.25 to the centre of the sun and reaches a normal area, which is oriented perpendicularly to the direction of radiation. We have revised the figures according to your suggestion and presented them as two separate figures. The PM1 figure shared the same X-axis labels as radiation figure, as explanted in figure caption, the X-axis was mixing layer height.

Fig S4: Please state the time period and explain x-axis (no axis label is provided)

Response: Thanks for the comment. The data was statistics of two years mixing layer height with variation of PM2.5, NOX and ozone. The X-label was added in the revised version.

Fig S5: Define Fb here. This figure does not appear relevant to the paper, and should be removed.

Response: Thanks for the comment. We have removed the figure.

Fig S5(2): Please correct the numbering of this and subsequent figures

Response: Thanks for the correction. We have numbered the figure correctly.

Fig S6: The increase in secondary aerosol is interesting here, but it would be more informative to show how the contribution of secondary aerosol to total aerosol changes here. However, the figure is not discussed in the paper, and should either be discussed or removed.

Response: Thanks for the comment. Inorganic aerosol like sulfate, nitrate and ammonium are can be considered as secondary aerosol. However, identifying primary and secondary of organic aerosol from a 2 years data needs more time with source apportionment method, and the current work cannot resolve it. We added more discussions in the revised version, as following: As shown in figure S7, the chemical concentration of NR-PM$_1$ increased significantly from 12.1 ug m$^{-3}$ to 56.4 ug m$^{-3}$ with the variation of MLH decreased from more than 1400 m to less than 200 m.

Fig S7: What period is this over? Again, this figure should be discussed or dropped.

Response: Thanks for the comment. The measurement was conducted from July 2009 to August of 2012, the figure has been removed.

Typos and minor issues

Please use European date order conventions (day, month, year) throughout. The x-axis labels in Fig 1 use US conventions, while Fig S5 in the supplement uses Chinese conventions.

Response: Thanks for the comment. We revised the Figure 1 and added time and year in X-label.

Title: episode -> episodes

line 107: in -> on (or at)

line 111: remove to

line 118: 220-400?

line 142: microgram unit missing

line 178: assign -> ascribe

line 184: bot ??

line 396-7: stand for -> represents

line 415: date -> data

Response: Thanks for the correction.

**Reviewer 2#**

This paper characterizes the interactions between atmospheric mixing layer dynamics and fine particulate matter pollution using long-term measurements of vertical distributions of PM2.5 and NOx, atmospheric mixing layer height, vertical meteorological parameters, energy flux, etc. in an urban site in Beijing. Based on the relationship between PM2.5 concentration, mixing layer height, solar radiation, and turbulent kinetic energy, the authors claimed that they found a feedback mechanism between mixing layer height and fine particulate matter pollution that could explain the rapid formation of severe haze pollution episodes in Beijing.

This work addresses an important topic that are of interest to many of the readers in atmospheric science community. However, many data presented in the paper are not thoroughly analyzed and discussed, and the evidence claimed by the authors are not strong enough to support their conclusion about the aerosol pollution-mixing layer development feedback mechanism. There are many important issues that need to be addressed before the publication of the paper in ACP can be considered.

Response: Thanks for the comment.

Major comments:

The authors claimed that the fine aerosols can reduce the solar radiation reaching the surface, resulting in a decrease in the turbulent kinetic energy (TKE) and a suppression of atmospheric mixing layer development, which further increase aerosol concentrations from direct emission and secondary formation (i.e., the feedback mechanism). However, they did not provide clear evidence that fine aerosols play a non-negligible role in regulating TKE and mixing layer heights. As shown in Fig 1, the TKE decreases dramatically from 8:00 to 20:00 on 21 November 2010, while aerosol loadings are pretty low during this period. This suggests that the variation of TKE is largely driven by non-haze related factors. Therefore, to claim the feedback mechanism, it is important to quantify to what extent fine aerosols can reduce or regulate the TKE and the development of the mixing layer in severe haze episodes.

Response: Thanks for the comment. The increasement of aerosol loading in the atmosphere can suppress vertical development of turbulent activity. However, other factors also will influence the variation of TKE, as the reviewer pointed out, the TKE decreases dramatically from 8:00 to 20:00 on 21 November 2010, while aerosol loadings are pretty low during this period. The dramatically variation of TKE was due to wind shear, as shown in Figure 1(a), the wind speed increased dramatically from 1m/s to 6m/s on the surface, so the wind shear could be occurred. As presented in figure s1, a cold front passed Beijing region on 21 November.

Other specific comments:

In the Introduction, the review of literatures is too brief. A summary of the current knowledge and remain issues regarding the interactions between boundary layer dynamics and aerosol pollution should be included, and the novelty of the present study should be clearly pointed out.

Response: Thanks for the comment. We added a summary of current knowledge and remain issues in our revised manuscript. Also, we pointed out the novelty of the present study. By using

filed measurements combined with model simulation, a positive Feedback between aerosol pollution, relative humidity and boundary layer was important in aerosol production, accumulation and severe haze formation in Beijing (Liu et al., 2018).    Wang et al., 2018 found that PBL schemes in their atmospheric chemistry models are not sufficient to describe the explosive growth of PM2.5 concentration in Beijing-Tianjin-Hebei region due to absence of an online calculation of aerosol-radiation feedback, and/or a deficient description of    extremely weak turbulent diffusion.

In this study, using unique measurements on the Beijing 325-meter-high meteorology tower, we show clear relationship between mixing layer height and turbulent kinetic energy at 140m observation platform. We also present direct evidence on the feedback that relates the decreasing mixed layer height with increasing particulate matter concentrations.

P4, Sect. 2.3 and 2.4. Please specify the altitude at which the measurements of O3, NOx, radiation, and aerosol chemical composition were performed.

Response: Thanks for the comment. We added statement that all these measurements were conducted in the IAP station.

Line 111. Remove "to", and "ratio" should be "rate".

Response: Thanks for the correction, we corrected.

Line 121-123. The HR-ToF-AMS was used to measure aerosol chemical composition. However, the data were not discussed in the paper, though a figure (Fig. S6) was included in the supplementary martial.

Response: Thanks for the comment. We added statements in our revised manuscript as following: As shown in figure S7, the chemical concentration of NR-PM$_1$ increased significantly from 12.1 ug m$^{-3}$ to 56.4 ug m$^{-3}$ with the variation of MLH decreased from more than 1400 m to less than 200 m.

Line 138-141 and Figs. S2 and S4. The decoupling of the 280-m platform from the other two lower ones for O3 was shown to be much smaller than that for PM2.5 and NOx. What is the reason for this difference?

Response: Thanks for the comment. The possible reason was that the concentrations of PM2.5 and NOx were higher near surface than that at higher altitude, while O3 concentration was higher at higher altitude (within 1000m), also O3 is a secondary product on the surface.

Line 171-173 and Fig. S3b. Was the PM1 measured by HR-ToF-AMS? If so, the authors should point out that measured PM1 mass concentrations do not include the refractory components such as soot and dust, whereas the PM2.5 concentrations include these components. In addition, compared to PM1-2.5, the origin of PM1 is generally more secondary. Therefore, the increase of PM1 concentration but decrease of its mass fraction in PM2.5 (as the decrease of mixing layer height) may offer insights into the contributions of primary emission and secondary formation to the haze pollution. This merits further discussions in the paper.

Response: Thanks for the comment. The mass concentration of PM1 was measured by Thermo TEOM 1400, which is the same instrument as PM2.5 measurement. In general, We agree your statement that PM1 is more secondary than PM1-2.5. However, during intensive haze period, the formation of PM1-2.5 is more significant than PM1, which is possible due to aqueous phase reactions and gas-particle partitioning of gases. For larger sized particles, these two process are more significant (Wang et al., 2015).

Line 184. "bot" should be "both".

Response: Thanks for the correction.

Some references in the reference list do not follow an alphabetical order.

Response: Thanks for the correction. We made the correct order in our revised version.

Reference

Tang, G., Zhang, J., Zhu, X., Song, T., Münkel, C., Hu, B., Schäfer, K., Liu, Z., Wang, L., Xin, J., Suppan, P., and Wang, Y.: Mixing layer height and its implications for air pollution over Beijing, China, Atmos. Chem. Phys., 16, 2459-2475, 10.5194/acp-16-2459-2016, 2016.

Liu, Q., Jia, X., Quan, J., Li, J., Li, X., Wu, Y., Chen, D., Wang, Z. and Liu, Y.: New positive feedback mechanism between boundary layer meteorology and secondary aerosol formation during severe haze events , doi:10.1038/s41598-018-24366-3, 2018.

Wang, Y. H., Z. R. Liu, J. K. Zhang, B. Hu, D. S. Ji, Y. C. Yu, and Y. S. Wang, Aerosol physicochemical properties and implications for visibility during an intense haze episode during winter in Beijing, Atmos. Chem. Phys., 15(6), 3205-3215, doi:10.5194/acp-15-3205-2015, 2015.

---

## Author Response (AR2)

Editor's comments

Dear Wang and co-authors,

Your revised manuscript has now been reviewed by two referees. Both referees still have some major concerns on the revised manuscript, particularly regarding observational evidence for the dynamical feedback between aerosol and mixing height.

Both referees still have major concerns about the main conclusion of your paper that you show observational evidence for a dynamical control of aerosol on mixing height. The observations clearly show evidence that aerosol concentration and mixing height are related. The evidence that this is controlled through a dynamical feedback of aerosol on mixing height, rather than purely the control of mixing height on aerosol concentration, is weaker.

Before the manuscript can be accepted for publication we would need to see a convincing argument for this dynamical feedback based on your observational analysis. Please include this additional analysis in your revised manuscript, in addition to any comments in the response to review.

I look forward to receiving your revised manuscript.

Regards,

Dominick Spracklen

Dear Dominick,

We thank you very much for the patient and encouragement on this resubmission.

In this revised version, we did major revisions about the manuscript. Firstly, modeling results of surface relative humidity, temperature and mixing layer height from WRF model were added. The model results clearly indicated a vital role of aerosol direct radiative forcing in development of mixing layer height (Line 188-Line 201).Then, we added more discussions on aerosol-boundary layer feedback mechanism carefully before we ascribe our measured phenomenon as aerosol-boundary layer feedback (Line 236-Line 246). Finally, we explained more about our fitting results about PM2.5-MLH relationship (Line 216-222). Please see details for more information.

On the basis of our observed results, we hope the revised manuscript has addressed your and reviewers' concerns. Meanwhile, we are happy to revise the manuscript further, if you have more comments and suggestions.

Best wishes,

Yonghong Wang on behalf of all co-authors

**A point to point response to the reviewers' comments**

We thank the two reviewers for their comments, and we do think their comments and suggestions improved our manuscript a lot. Here are the points to points responses (in blue colored), accordingly, we also revised manuscript.

Report#1

> The revised manuscript has addressed many of my minor concerns, but my principal criticism has not been addressed adequately yet. The manuscript provides valuable observational evidence for an inverse relationship between aerosol and mixing height, but this is not sufficient to demonstrate the presence of a feedback without further supporting evidence.

In their response, the authors reassert the existence of the dynamical feedback. The theoretical basis for this is sound, but the potential strength of this study is in providing observational evidence for it. The authors are too quick to demonstrate that the observations are consistent with the theory, and neglect the much more valuable goal of testing the theory based on the observations. This is evident in their response to my request for justification for fitting an exponential curve rather than a reciprocal. The authors present an interesting analysis in the response to reviewers, but have not provided a justification or made any changes to the manuscript. What relationship fits the observations best, and what implications does this have for the theory about a feedback?

Response: We thank you for the very important comments. Aerosol-boundary interaction and related feedback mechanisms have been subjected intensive studies due to the vital role in air pollution. Most of these studies were conducted by model simulation combined with some measurements data. In our study, we presented vertical measurements of aerosol concentration combined with mixing layer height with three years of measurement. The two measurements were connected by turbulent kinetic energy (TKE) obtained from 140 m observation plate to demonstrate aerosol-boundary layer feedback. In our opinion, the novelty of the work is the unique measurements with such a long date sets. Also, to the best of our knowledge, this is the first time we show a negative correlation about mixing layer height and TKE observed above surface, (as shown in Figure 5 ), which is a key process in aerosol -boundary layer feedback loop. We were able to quantify the feedback by compare increased $PM_{2.5}$ with the increased amounts of NOx. According to your comments, we did major revision on the manuscript: first, we added WRF model results during the intensive haze periods as we shown in Figure 1 to show the importance of aerosol in developing of mixing layer height. Basically, we conducted control experiment and test experiment by considering or without considering aerosol direct radiative forcing in the model. The results clearly showed that the consideration of aerosol direct radiative forcing into the model lead to decreased surface air temperature, increased relative humidity and suppressed development of mixing layer height in urban Beijing (Line 188- Line202). The second major revision was that we discussed aerosol-boundary layer feedback mechanism carefully before we ascribe observed phenomenon as aerosolboundary layer feedback (Line 236- line246).

Finally, we got an exponential curve rather than a reciprocal in our Figure 4 was that the reciprocal fitting overestimated the $PM_{2.5}$ concentration when the mixing layer height was very low compared to the exponential fitting function (Figure. 4), which also indicated that a much higher $PM_{2.5}$ concentration was needed in order to obtain a very low mixing layer height without the positive feedback. We revised this part and added more statements. (Line 216-Line 222)

The conclusions of the paper remain weak, as noted in my original review, but have not been altered in the revised version.

Response: we added more statements in our conclusion about the feedback mechanism as the reviewer commented. (Line 270-Line 277).

As I commented in my original review, the final sentence of the abstract needs revision: most good air quality models have included this feedback for many years (albeit without strong observational support, which this study could provide) so it is too late to "suggest that the feedback mechanism should be considered". The study just reconfirms that it should be considered. The changes made to the final paragraph of the introduction (l.80-85) have improved it and now more accurately summarise the findings of the study, but the abstract does not reflect this yet.

Response: We thank you for the comment, we revised the statement in our abstract. (Line 45,46)

L.183: As noted in my point above, the fitting of an exponential to Figure 3 still isn't explained. The new text at L.195 is not adequate to explain it. No reasoning or support is provided for ascribing the observed behaviour to a positive feedback (L.200-201).

Response: Thanks for the comment. We added more explanations about this part. (Line216-Line 222 )

The English language is generally reasonable but needs further polishing before the paper is suitable for publication.

Response: We revised the manuscript carefully in this version, we hope it can meet the scope.

For an English Language journal it would be appropriate to replace the "D" and "G" in Figure S1 with "L" and "H" (the figure should be in English, not Chinese).

Response: Thanks for the reminder, we revised these symbols in the revised version.

A number of the references remain incomplete, e.g., missing journal for Wang et al. 2018, missing details/date for Petaja et al. and Ding et al., and formatting errors for several other references.

Response: Thanks for the corrections and we revised them in the current version.

The English in the title has been corrected, but in practice the title does not accurately reflect the topic or findings of the paper.

Response: We revised the title as 'Rapid formation of intense haze episodes via aerosol-boundary layer feedback in Beijing', which we hope it suitable for the paper.

Report#2

The main conclusion of this study is the two-way feedback between aerosol pollution and mixing layer height. However, neither the reviewer #1 nor I believed that the authors had provided extensive evidence to support this feedback mechanism in their previous manuscript. Unfortunately, the authors did not add any new data or appropriate analysis to validate their conclusion in the revised manuscript. Therefore, I cannot recommend the publication of this manuscript in its current version in ACP.

Response: Thank you for your comments, which motivated us to revise the manuscript considerable.

In this revised version, we added modeled results of surface relative humidity, temperature and mixing layer height from WRF model. The model results clearly indicated the vital role of aerosol direct radiative forcing in development of mixing layer height (Line 188-Line 201). Secondly, we added more discussions carefully on aerosol-boundary layer feedback mechanism before we ascribe our measured phenomenon as aerosol-boundary layer feedback (Line 236-Line 246). Thirdly, we explained more about our fitting results about PM2.5-MLH relationship (Line 216-222).

Finally but the most importantly, to the best of our knowledge, this is the first time that three years of the vertical air pollutants measurements were presented, combined with mixing layer height information and turbulent kinetic energy results, we do think our results provided here benefit current information of boundary layer- aerosol feedbacks in highly polluted urban cities.